# Fixes That Fail: Self-Defeating Improvements in Machine-Learning Systems

**Ruihan Wu**[*]  
Cornell University  
rw565@cornell.edu

**Chuan Guo**      **Awni Hannun**      **Laurens van der Maaten**  
Facebook AI Research  
{chuanguo,awni,lvdmaaten}@fb.com

## Abstract

Machine-learning systems such as self-driving cars or virtual assistants are composed of a large number of machine-learning models that recognize image content, transcribe speech, analyze natural language, infer preferences, rank options, *etc*. Models in these systems are often developed and trained independently, which raises an obvious concern: *Can improving a machine-learning model make the overall system worse?* We answer this question affirmatively by showing that improving a model can deteriorate the performance of downstream models, even after those downstream models are retrained. Such *self-defeating improvements* are the result of entanglement between the models in the system. We perform an error decomposition of systems with multiple machine-learning models, which sheds light on the types of errors that can lead to self-defeating improvements. We also present the results of experiments which show that self-defeating improvements emerge in a realistic stereo-based detection system for cars and pedestrians.

## 1 Introduction

Progress in machine learning has allowed us to develop increasingly sophisticated artificially intelligent systems, including self-driving cars, virtual assistants, and complex robots. These systems generally contain a large number of machine-learning models that are trained to perform modular tasks such as recognizing image or video content, transcribing speech, analyzing natural language, eliciting user preferences, ranking options to be presented to a user, *etc*. The models feed each other information: for example, a pedestrian detector may use the output of a depth-estimation model as input. Indeed, machine-learning systems can be interpreted as directed acyclic graphs in which each vertex corresponds to a model, and models feed each other information over the edges in the graph.

In practice, various models in this graph are often developed and trained independently [22]. This modularization tends to lead to more usable APIs but may also be motivated by other practical constraints. For example, some of the models in the system may be developed by different teams (some models in the system may be developed by a cloud provider); models may be retrained and deployed at a very different cadence (personalization models may be updated very regularly but image-recognition models may not); new downstream models may be added continuously in the graph (user-specific models may need to be created every time a user signs up for a service); or models may be non-differentiable (gradient-boosted decision trees are commonly used for selection of discrete features). This can make backpropagation through the models in the graph infeasible or highly undesirable in many practical settings.

The scenario described above leads to an obvious potential concern: *Can improving a machine-learning model make the machine-learning system worse?* We answer this question affirmatively by showing how improvements in an upstream model can deteriorate the performance of downstream models, *even if all the downstream models are retrained* after updating the upstream model. Such

---

[*]Work performed during internship at Facebook AI Research.

35th Conference on Neural Information Processing Systems (NeurIPS 2021).

*self-defeating improvements* are caused by entanglement between the models in the system. To better understand this phenomenon, we perform an error decomposition of simple machine-learning systems. Our decomposition sheds light on the different types of errors that can lead to self-defeating improvements: we provide illustrations of each error type. We also show that self-defeating improvements arise in a realistic system that detects cars and pedestrians in stereo images.

Our study opens up a plethora of new research questions that, to the best of our knowledge, have not yet been studied in-depth in the machine-learning community. We hope that our study will encourage the community to move beyond the study of machine-learning models in isolation, and to study machine-learning systems more holistically instead.

## 2    Problem Setting

We model a machine-learning *system* as a static directed acyclic graph (DAG) of machine-learning *models*, $G = (\mathcal{V}, \mathcal{E})$, where each vertex $v \in \mathcal{V}$ corresponds to a machine-learning model and each edge $(v, w) \in \mathcal{E}$ represents the output of model $v$ being used as input into model $w$. For example, vertex $v$ could be a depth-estimation model and vertex $w$ a pedestrian-detection model that operates on the depth estimates produced by vertex $v$. A model is *upstream* of $v$ if it is an ancestor of $v$. A *downstream* model of $v$ is a descendant of $v$. An example of a machine-learning system with three models is shown in Figure 1 (we study this model in Section 3.2).

Each vertex in the machine-learning system $G$ corresponds to a model function, $v(\cdot)$, that operates on both data and outputs from its upstream models. Denote the output from model $v$ by $f_v(\mathbf{x})$. The output of a source model $v$ is $f_v(\mathbf{x}) = v(\mathbf{x}^{(v)})$, where $\mathbf{x}^{(v)}$ represents the part of input $\mathbf{x}$ that is relevant to model $v$. The output of a non-source model $w$ is given by $f_w(\mathbf{x}) = w\left(\left[\mathbf{x}^{(w)}, f_v(\mathbf{x}) : v \in \mathcal{P}_w\right]\right)$, where $\mathcal{P}_w = \{v : (v, w) \in \mathcal{E}\}$ are the parents of $w$.

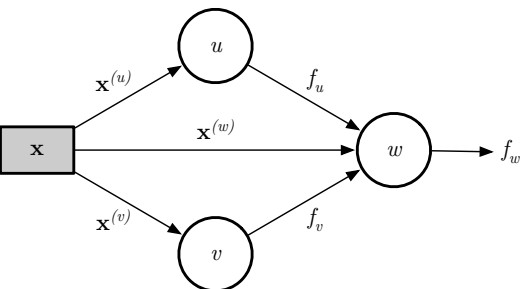

**Training.**    To train model $v$, we assume access to a training set $\mathcal{D}_v = \{(\mathbf{x}_1, \mathbf{y}_1), \dots, (\mathbf{x}_{N_v}, \mathbf{y}_{N_v})\}$ with $N_v$ examples $\mathbf{x}_n \in \mathcal{X}$ and corresponding targets $\mathbf{y}_n \in \mathcal{Y}_v$. For each model, we also assume a training algorithm $A_v(\mathcal{D}_v)$ that selects a model $v \in \mathcal{H}_v$ from hypothesis set $\mathcal{H}_v$ based on the training set. The learning algorithm, $A_v$, the hypothesis set, $\mathcal{H}_v$, and the training set, $\mathcal{D}_v$, are fixed during

Figure 1: Example of a machine-learning system with three models, $\mathcal{V} = \{u, v, w\}$, and two edges, $\mathcal{E} = \{(u, w), (v, w)\}$. The system receives an example $\mathbf{x}$ as input, of which parts $\mathbf{x}^{(u)}$, $\mathbf{x}^{(v)}$, and $\mathbf{x}^{(w)}$ are fed into $u$, $v$, and $w$, respectively. Model $w$'s input is concatenated with the outputs of its parents, $f_u(\mathbf{x})$ and $f_v(\mathbf{x})$.

training but they may change each time model $v$ is re-trained or updated. The data space $\mathcal{X}$, the target space $\mathcal{Y}_v$, and the way in which inputs $\mathbf{x}^{(v)}$ are obtained do not change between model updates.

We assume that models in $G$ may be trained *separately* rather than jointly, that is, the learning signal obtained from training a downstream model, $v$, may not be backpropagated into its upstream dependencies, $w \in \mathcal{P}_v$. This assumption is in line with constraints that commonly arise in real-world machine learning systems; see our motivation below. Note that this assumption also implies that a downstream model cannot be trained before all its upstream models are trained.

**Evaluation.** To be able to evaluate the performance of the task at $v$, we assume access to a *fixed* test set $\bar{\mathcal{D}}_v$ that is defined analogously to the training set: it contains $M_v$ test examples $\bar{\mathbf{x}}_m \in \mathcal{X}$ and corresponding targets $\bar{\mathbf{y}}_m \in \mathcal{Y}_v$. In contrast to the training set, the test set is fixed and does not change when the model is updated. However, we note that the input distribution into a non-root model is not only governed by the test set $\bar{\mathcal{D}}_v$ but also by upstream models: if an upstream model changes, the inputs into a downstream model may change as well even though the test set is fixed.

We also assume access to a *test loss* function $\ell_v(v, \bar{\mathcal{D}}_v)$ for each model $v$, for example, classification error (we assume lower is better). Akin to the test set $\bar{\mathcal{D}}_v$, all test loss functions $\ell_v$ are fixed. Hence, we always measure the generalization ability of a model, $v$, in the exact same way. The test loss function may not have trainable parameters, that is, it must be a "true" loss function.

**Updating models.** The machine-learning system we defined can be improved by updating a model. In such an update, an individual model $v$ is replaced by an alternative model $v'$. We can determine if such a model update constitutes an *improvement* by evaluating whether or not its test loss decreases: $\ell_v(v', \bar{\mathcal{D}}_v) \leq \ell_v(v, \bar{\mathcal{D}}_v)$. When $v$ is not a leaf in $G$, the update to $v'$ may affect the test loss of downstream models as well. In practice, the improvement due to $v'$ often also improves downstream models (*e.g.*, [19]), but this is not guaranteed. We define a *self-defeating improvement* as a situation in which replacing $v$ by $v'$ improves that model but deteriorates at least one of the downstream models $w$, even after all the downstream models are updated (*i.e.*, re-trained using the same learning algorithm) to account for the changes incurred by $v'$.

**Definition 2.1** (Self-defeating improvement). Denoting the set of all downstream models (descendants) of $v$ in $G$ as $\mathcal{C}_v$, a *self-defeating improvement* arises when:

$$\exists w \in \mathcal{C}_v : \ell_v(v', \bar{\mathcal{D}}_v) \leq \ell_v(v, \bar{\mathcal{D}}_v) \;\not\Longrightarrow\; \ell_w(w', \bar{\mathcal{D}}_w) \leq \ell_w(w, \bar{\mathcal{D}}_w)), \tag{1}$$

where $w'$ represents a new version of model $w$ that was trained after model $v$ was updated to $v'$ to account for the input distribution change that the update of $v$ incurs on $w$.

**Motivation of problem setting.** In the problem setting described above, we have made two assumptions: (1) models may be (re-)trained separately rather than jointly and (2) the training algorithm, hypothesis set, and training data may change between updates of the model.[2] Both assumptions are motivated by the practice of developing large machine-learning systems that may comprise thousands of models [22, 28]. Joint model training is often infeasible in such systems because:

- Some of the models may have been developed by cloud providers [34] and cannot be changed.
- Backpropagation may be technically infeasible or inefficient, *e.g.*, when models are implemented in incompatible learning frameworks or when their training data live in different data centers.
- Some models may be re-trained or updated more often than others. For example, personalization models may be updated every few hours to adapt to changes in the data distribution, but re-training large language models [8] at that same cadence is not very practical.
- Upstream models may have hundreds or even thousands of downstream dependencies, complicating appropriate weighting of the downstream learning signals that multi-task learning [10] requires.
- Some models may be non-differentiable, *e.g.*, gradient-boosted decision trees are commonly used for feature selection in large sets of features.

We assume that changes in the training algorithm, hypothesis set, and training data may occur because model owners (for example, cloud providers) constantly seek to train and deploy improved versions of their models with the ultimate goal of improving the system(s) in which the models are used.

## 3 Understanding Self-Defeating Improvements via Error Decompositions

Traditional software design relies heavily on *modules*: independent and interchangeable components that are combined to form complex software systems [7, §7.4]. Well-designed modules provide an explicit specification of their inputs, outputs, and functionality. In traditional software engineering, good modularization allows developers to modify modules independently because the effects of those modifications on the rest of the system are relatively predictable.

Machine-learned models do not resemble traditional software modules because they lack an explicit specification of their functionality [14]. As a result, there is often a significant degree of *entanglement* between different models in the system [22]. This makes the system-wide effects of model modifications unpredictable, and can lead to situations where improving a model does not improve the entire system. To understand such self-defeating improvements, we study Bayes error decompositions of simple machine-learning systems. Specifically, we study a system with two models in Section 3.1 and a system with three models in Section 3.2.

### 3.1 Error Decomposition of System with Two Models

We adapt standard Bayes error decomposition [6, 33] to study a simple system that has two vertices, $\mathcal{V} = \{v, w\}$, and a dependency between upstream model $v$ and downstream model $w$, that is, $\mathcal{E} = \{(v, w)\}$. We denote the Bayes-optimal downstream model by $w^*$; the optimal downstream

---

[2]The training data is fixed during training, *i.e.*, we do not consider online learning settings in our setting.

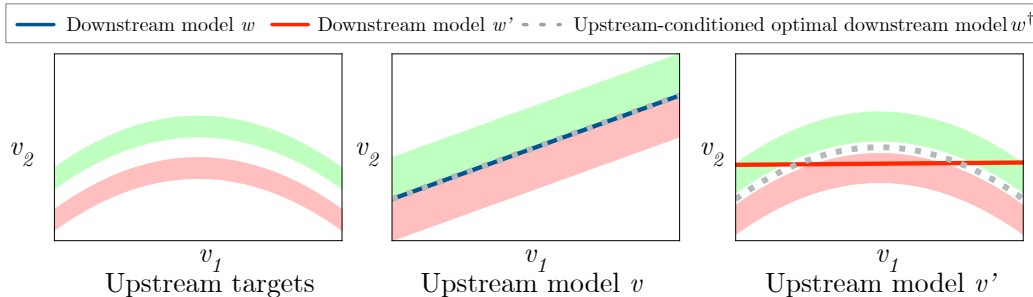

Figure 2: Illustration of a self-defeating improvement due to an increase in the downstream approximation error. Upstream model $v$ predicts a 2D point for each example. Downstream model $w$ is a linear classifier separating the points into two classes (red and green). **Left:** Ground-truth target distribution for the upstream model. **Middle:** Point predictions by upstream model $v$, the optimal $v$-conditioned downstream classifier $w^\dagger$, and the downstream classifier $w$ learned. **Right:** Point predictions by improved upstream model $v'$, the optimal $v'$-conditioned downstream classifier $w^\dagger$, and the downstream classifier $w'$ learned. Note how the predictions of $v'$ better match the ground-truth, but $w'$ cannot exploit the improvement because it is restricted to be linear.

model conditioned on an upstream model by $w^\dagger$; and the optimal upstream-conditional downstream model in the hypothesis set $\mathcal{H}_w$ by $w^\dagger_{\mathcal{H}_w}$. Using these definitions, we can decompose the *downstream risk* of model $w$, given upstream model $v$, as follows:

$$\mathbb{E}[\ell_w(w \circ v) - \ell_w(w^*)] =$$
$$\underbrace{\mathbb{E}[\ell_w(w^\dagger \circ v) - \ell_w(w^*)]}_{\text{upstream error}} + \underbrace{\mathbb{E}[\ell_w(w^\dagger_{\mathcal{H}_w} \circ v) - \ell_w(w^\dagger \circ v)]}_{\text{downstream approximation error}} + \underbrace{\mathbb{E}[\ell_w(w \circ v) - \ell_w(w^\dagger_{\mathcal{H}_w} \circ v)]}_{\text{downstream estimation error}}, \quad (2)$$

where $\circ$ denotes function composition, and the expectations are under the data distribution. The error decomposition is similar to standard decompositions [6, 33] but contains three terms instead of two.[3] Specifically, the *upstream error* does not arise in prior error decompositions, and the *downstream approximation* and *downstream estimation* errors differ from the standard decomposition.

Barring variance in the error estimate due to the finite size of test sets $\bar{\mathcal{D}}_w$ (which is an issue that can arise in any model-selection problem), a self-defeating improvement occurs when a model update from $(v, w)$ to $(v', w')$ reduces the upstream risk but increases the downstream risk. An increase in the downstream risk implies that at least one of the three errors terms in the composition above must have increased. We describe how each of these terms may increase due to a model update:

- **Upstream error** measures the error that is due to the upstream model $v$ not being part of the Bayes-optimal solution $w^*$. It increases when an improvement in the upstream loss does not translate in a reduction in Bayes error of the downstream model, *i.e.*, when: $\ell_w(w^\dagger \circ v') > \ell_w(w^\dagger \circ v)$. An upstream error increase can happen due the *loss mismatch*: a setting in which the test loss functions, $\ell$, of the upstream and downstream model optimize for different things. For example, the upstream test loss function may not penalize errors that need to be avoided in order for the downstream test loss to be minimized, or it may penalize errors that are irrelevant to the downstream model.

  Alternatively, the upstream error may increase due to *distribution mismatch*: situations in which the upstream loss focuses on parts of the data-target space $\mathcal{X} \times \mathcal{Y}_v$ that are unimportant for the downstream model (and vice versa). For example, suppose the upstream model $v$ is an image-recognition model that aims to learn a feature representation that separates cats from dogs, and $f_v(\mathbf{x})$ is a feature representation obtained from that model. A downstream model, $w$, that distinguishes different dog breeds based on $f_v$ may deteriorate when the improvement of model $v$ collapses all representations of dog images in $f_v$. Examples of this were observed in, *e.g.*, [19].

- **Downstream approximation error** measures the error due to the optimal $w^\dagger$ not being in the hypothesis set $\mathcal{H}_w$. The approximation error increases when the downstream model, $w$, is unable to exploit improvements in the upstream model, $v$, because exploiting those improvements would require selection of a model that is not in that hypothesis set.

---

[3]For brevity, we do not include optimization error of [5] in our Bayes error decomposition.

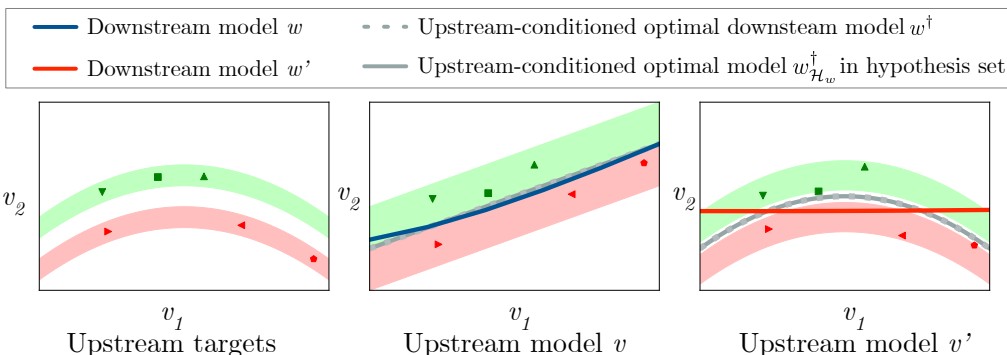

Figure 3: Illustration of a self-defeating improvement due to an increase in the downstream estimation error. Upstream model $v$ predicts a 2D point for each example. Downstream model $w$ is a quadratic classifier separating the points into two classes (red and green). **Left:** Ground-truth target samples and distribution for the upstream model. **Middle:** Point predictions by upstream model $v$, the corresponding optimal $v$-conditioned downstream model in $\mathcal{H}_w$, $w^\dagger_{\mathcal{H}_w}$, and the downstream classifier $w$ learned based on the training examples. **Right:** Point predictions by improved upstream model $v'$, the corresponding optimal $v'$-conditioned downstream model in $\mathcal{H}_w$, $w^\dagger_{\mathcal{H}_w}$, and the downstream classifier $w'$ learned based on the training examples. Note how the predictions of $v'$ better match the ground-truth, but $w'$ deteriorates because it only receives $N_w = 6$ training examples.

Figure 2 illustrates how this situation can lead to a self-defeating improvement. In the illustration, the upstream model, $v$, predicts a 2D position for each data point. The downstream model, $w$, separates examples into the positive (green color) and negative (red color) class based on the prediction of $v$ using a linear classifier. The predictions of the improved upstream model, $v'$, better match the ground-truth targets: the predictions of $v'$ (right plot) match the ground-truth targets (left plot) more closely than those of $v$ (middle plot). However, the re-trained downstream model, $w'$, deteriorates because the resulting classification problem is more non-linear: the downstream model cannot capitalize on the upstream improvement because it is linear. The resulting increase in approximation error leads to the self-defeating improvement in Figure 2.

- **Downstream estimation error** measures the error due to the model training being performed on a finite data sample with an imperfect optimizer, which makes finding $w^\dagger_{\mathcal{H}_w}$ difficult in practice. The estimation increases, for example, when the upstream model improves but the downstream model requires more than $N_w$ training samples to capitalize on this improvement.

Figure 3 shows an example of a self-defeating improvement caused by an increase of the downstream estimation error. As before, the upstream model, $v$, predicts a 2D position for each data point in the example. The downstream model $w$ is selected from the set, $\mathcal{H}_w$, of quadratic classifiers. It is tasked with performing binary classification into a positive class (green color) and negative class (red color) based on the 2D positions predicted by the upstream model. To train the binary classifier, the downstream model $w$ (and $w'$) is provided with $N_w = 6$ labeled training examples (the green and red markers in the plot). In the example, the upstream model $v'$ performs better than its original counterpart $v$: the predictions of $v'$ (right plot) match the ground-truth targets (left plot) more closely than those of $v$ (middle plot). However, the upstream improvement hampers the downstream model even though the optimal downstream model $w^\dagger$ is in the hypothesis set $\mathcal{H}_w$ both before and after the upstream improvement. After the upstream improvement, the optimal downstream model that can be selected from the hypothesis set, $w^\dagger_{\mathcal{H}_w}$, is more complex, which makes it harder to find it based on the finite number of $N_w$ training examples. This results in an increase in the estimation error, which causes the self-defeating improvement in Figure 3.

## 3.2   Error Decomposition of System with Two Upstream Models

More complex types of self-defeating improvements may arise when a downstream model depends on multiple upstream models that are themselves entangled. Consider the system of Figure 1 that has three models, $\mathcal{V} = \{u, v, w\}$, and dependencies between upstream models $u$ and $v$ and downstream model $w$, that is, $\mathcal{E} = \{(u, w), (v, w)\}$. The error decomposition for this system is

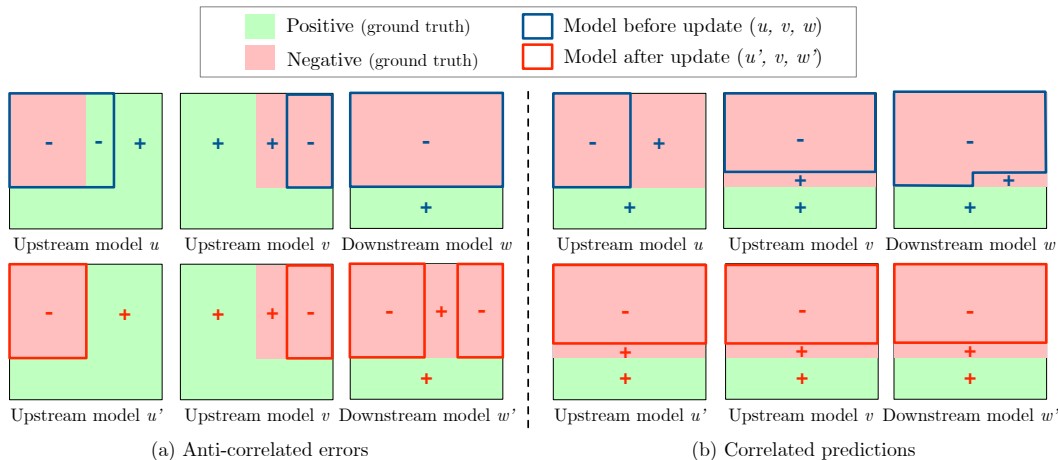

(a) Anti-correlated errors        (b) Correlated predictions

Figure 4: Illustrations of self-defeating improvement due to an increase in the upstream compatibility error. Green areas should be labeled as positive; red areas as negative. **Left** (a and b): Original upstream model $u$ (top; blue line and $+, -$ symbols) and its improved version $u'$ (bottom; red). **Middle** (a and b): Upstream model $v$. **Right** (a and b): Original downstream model $w$ (top) and its improved version $w'$ (bottom). In (a), the errors of $u$ and $v$ are anti-correlated. As a result, the improved upstream model $u'$ negatively impacts the re-trained downstream model $w'$. In (b), the improved upstream model $u'$ is identical to the other upstream model $v$. As a result, $w'$ looses the additional information previously provided by $u$, negatively impacting its performance.

similar to Equation 2, but we can further decompose the upstream error. Denoting the Bayes-optimal downstream model by $w^*$, the optimal model given upstream models $u$ and $v$ by $w_{u,v}^\dagger$, and the optimal upstream model $v$ given upstream model $u$ by $v_u^\dagger$, we decompose the upstream error as follows:

$$\underbrace{\mathbb{E}[\ell_w(w_{u,v}^\dagger \circ (u,v)) - \ell_w(w^*)]}_{\text{upstream error}} =$$

$$\underbrace{\mathbb{E}[\ell_w(w_{u,v}^\dagger \circ (u,v)) - \ell_w(w_{u,v}^\dagger \circ (u,v_u^\dagger))]}_{\text{upstream compatibility error}} + \underbrace{\mathbb{E}[\ell_w(w_{u,v}^\dagger \circ (u,v_u^\dagger)) - \ell_w(w^*)]}_{\text{excess upstream error}}. \quad (3)$$

The **excess upstream error** is similar to the upstream error in Equation 2: upstream model $u$ may be suboptimal for the downstream task, for example, because of loss mismatch or distribution mismatch. The key observation in the error decomposition of a system with two upstream models is that the optimal upstream model $v$ is a function of upstream model $u$ (and vice versa). The **upstream compatibility error** captures the error due to upstream model $v$ not being identical to the optimal $v_u^\dagger$. A self-defeating improvement can occur when we update upstream model $u$ to $u'$ because it may be that $v_u^\dagger \neq v_{u'}^\dagger$, which can cause the upstream compatibility error to increase.

We provide two examples of this in Figure 4. The first example (left pane) shows a self-defeating improvement due to upstream models $u$ and $v$ making *anti-correlated errors* that cancel each other out. The second example (right pane) is a self-defeating improvement due to $u'$ making more *correlated predictions* with $v$. We note that, in both examples, the optimal $v$ depends on $u$ and the self-defeating improvement arises because of an increase in the upstream compatibility error.

In the examples in Figure 4, all three models aim to distinguish two classes: green (positive class) and red (negative class). Upstream models $u$ and $v$ do so based the $(x, y)$-location of points in the 2D plane. The downstream model operates on the (hard) outputs of the two upstream models. In Figure 4(a), the original upstream models $u$ and $v$ make errors that are anti-correlated. The original downstream model $w$ exploits this anti-correlation to make perfect predictions. When upstream model $u$ is replaced by an improved model $u'$ that makes no errors, a self-defeating improvement arises: downstream model $w$ no longer receives the information it needs to separate both classes perfectly. In Figure 4(b), upstream model $u$ is improved to $u'$, which makes the exact same predictions as the other upstream model $v$. As a result, downstream model $w'$ no longer has access to the complementary information that $u$ was providing in addition to $v$, producing the self-defeating improvement.

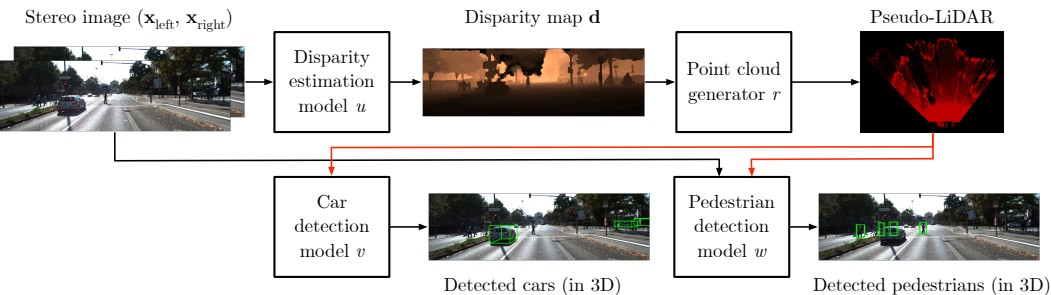

Figure 5: Overview of the car and pedestrian detection system used in our case study in Section 4. The system consists of three models: (1) a disparity-estimation model $u$ that generates a pseudo-LiDAR representation of the environment, (2) a car-detection model $v$ that operates on 3D point clouds, and (3) a pedestrian-detection model $w$ that performs 3D pedestrian detection.

While the examples in Figure 4 may seem contrived, anti-correlated errors and correlated predictions can arise in practice when there are dependencies in the targets used to train different models. For example, suppose the upstream models are trained to predict ad clicks (*will a user click on an ad?*) and ad conversions (*will a user purchase the product being advertised?*), respectively. Because conversion can only happen after a click happens, there are strong dependencies between the targets used to train both models, which may introduce correlations between their predictions.

## 4   Case Study: Pseudo-LiDAR for Detection

We perform a case study on self-defeating improvements in a system that may be found in self-driving cars. In both cases, the system uses a pseudo-LiDAR [37] as the basis for 3D detection of cars and pedestrians. Figure 5 gives an overview of the models in the system:

- A disparity-estimation model, $u$, predicts a map $\mathbf{d}$ with disparities corresponding to every pixel in the left image of a stereo image (a pair of rectified images from two horizontally aligned cameras).
- A point cloud generator, $r$, uses disparity map $\mathbf{d}$ to create a pseudo-LiDAR representation of the 3D environment. The point cloud generator, $r$, is a simple non-learnable module [37].
- The stereo image and disparity map are input into a model, $v$, that performs 3D detection of cars.
- The same inputs are used in a separate model, $w$, that aims to perform 3D pedestrian detection.

### 4.1   System Design

The disparity-estimation and detection models used in our case study are described below.

**Disparity estimation.** To perform disparity estimation, we adopt the PSMNet model of Chang and Chen [11]. The model receives a stereo image, $(\mathbf{x}_{\text{left}}, \mathbf{x}_{\text{right}})$, as input and aims to compute a disparity map, $\mathbf{d} = u(\mathbf{x}_{\text{left}}, \mathbf{x}_{\text{right}})$, that contains a disparity estimate for each pixel in image $\mathbf{x}_{\text{left}}$. We experiment with two training loss functions: (1) the depth mean absolute error and (2) the disparity mean absolute error (MAE). Given a ground-truth disparity map, $\mathbf{y}$, and a function that maps disparities to depths, $g(\mathbf{d}) = \frac{C}{\mathbf{d}}$, for some camera constant $C$, the disparity MAE is proportional to $\|\mathbf{d} - \mathbf{y}\|_1$ and the depth MAE is proportional to $\|g(\mathbf{d}) - g(\mathbf{y})\|_1$. To evaluate the quality of model $u$, we measure disparity MAE: $\ell_u(u, \bar{\mathcal{D}}_u) = \frac{1}{M} \sum_{m=1}^{M} \|\mathbf{d} - \mathbf{y}\|_1$. We expect that a model, $u'$, trained to minimize disparity

|  |  | Car detection | | Pedestrian detection | |
|---|---|---|---|---|---|
|  |  | **P-RCNN** | **F-PTNET** | **P-RCNN** | **F-PTNET** |
| $AP_{3D}$ | $u$ | **39.91** | **33.64** | **34.42** | 43.02 |
|  | $u'$ | 38.75 | 32.78 | 27.64 | **43.10** |
|  | $y$ | 68.06 | 55.46 | 55.19 | 62.16 |
| $AP_{BEV}$ | $u$ | **50.34** | **43.79** | **38.51** | 50.49 |
|  | $u'$ | 49.30 | 42.77 | 31.47 | 48.29 |
|  | $y$ | 78.43 | 67.35 | 63.00 | 65.92 |

Table 1: Average precision of car detection and pedestrian detection measured for 3D box view ($AP_{3D}$) and 2D birds-eye view ($AP_{BEV}$). Higher is better. Results are shown for baseline disparity-estimation model $u$ and an improved disparity-estimation model $u'$ that is trained using a different loss. The AP of oracle disparity-estimation model $y$ is shown for reference. Improving the disparity-estimation model leads to a self-defeating improvement in both the car detection model and the pedestrian detection model.

MAE will provide better disparity estimates (per test loss $\ell_u$) than a model, $u$, that is trained to minimize depth MAE. This has downstream effects on performance of the detection models.

**Car and pedestrian detection.** We experiment with two different models for performing detection of cars and pedestrians, *viz.*, the Point-RCNN model (P-RCNN; Shi et al. [30]) and the Frustum PointNet detection model (F-PTNET; Qi et al. [26]). Both models take stereo image $(\mathbf{x}_{\mathrm{left}}, \mathbf{x}_{\mathrm{right}})$ and point cloud $r(\mathbf{d})$ as input. Before computing $r(\mathbf{d})$, we perform winsorization on the prediction $u(\mathbf{x}_{\mathrm{left}}, \mathbf{x}_{\mathrm{right}})$: we remove all points that are higher than 1 meter in the point cloud (where the camera position is the origin). Both 3D car detection model and 3D pedestrian detection model are trained to detect their target objects at any distance from the camera.

Following Geiger et al. [16], we evaluate the test loss of car-detection model, $\ell_v$, using the negative average precision (AP) for an intersection-over-union (IoU) of $0.7$. The test loss of the pedestrian-detection model, $\ell_w$, is the negative AP for an IoU of $0.5$. The pedestrian-detection test-loss is only evaluated on pedestrians whose distance to the camera is $\leq 20$ meters. We measure both the car-detection and the pedestrian-detection test losses for both the 3D box view ($-\mathrm{AP}_{\mathrm{3D}}$) and the 2D bird-eye view ($-\mathrm{AP}_{\mathrm{BEV}}$); see Geiger et al. [16] for details on the definition of the test-loss functions.

## 4.2 Experiments

We evaluate our system on the KITTI dataset [16, CC BY-NC-SA 3.0], using the training-validation split of [12]. Our baseline disparity-estimation model, $u$, that is trained to minimize depth MAE obtains a test loss of 1.28 (see Table 2). The improved version of that model, $u'$, is trained to minimize disparity MAE and obtains a test loss of 1.21, confirming the model improvement.

| Range | 0-max | 0-20 | 20-40 | 40-max |
|---|---|---|---|---|
| **Model $u$** | 1.28 | 1.32 | 1.12 | 1.11 |
| **Model $u'$** | 1.21 | 1.21 | 1.20 | 1.20 |

Table 2: Disparity MAE test loss of disparity-estimation models $u$ (trained to minimize depth MAE) and $u'$ (trained to minimize disparity MAE), split out per range of ground-truth depth values (in meters). Model $u$ better predicts the disparity of nearby points, but $u'$ works better on distant points.

Table 1 presents the test losses of the downstream models for car and pedestrian detection, for both the baseline upstream model $u$ and the improved model $u'$. We observe that the improvement in disparity estimation leads to self-defeating improvements on both the car-detection and the pedestrian-detection tasks. The AP$_{\mathrm{3D}}$ of the car detector drops from 39.91 to 38.75 (from 50.34 to 49.30 in AP$_{\mathrm{BEV}}$). For pedestrian detection, AP$_{\mathrm{3D}}$ is unchanged but AP$_{\mathrm{BEV}}$ drops from 50.49 to 48.29. Interestingly, these self-defeating improvements are due to increases in different error terms.

**Self-defeating improvement in car detection.** The self-defeating improvement observed for car detection is likely due to an increase in the upstream error. A disparity-estimation model that is trained to minimize depth MAE (*i.e.*, model $u$) focuses on making sure that *small* disparity values (large depth values), are predicted correctly. By contrast, a model trained to minimize disparity MAE (model $u'$) aims to predict *large* disparity values (small depth values) correctly, sacrificing accuracy in predictions of large depth values (see Table 2). This negatively affects the car detector because most cars tend to be relatively far away. In other words, the loss function that improves model $u$ deteriorates the downstream model because it focuses on errors that are less relevant to that model, which increases the upstream error.

**Self-defeating improvement in pedestrian detection.** Different from car detection, the self-defeating improvement in pedestrian detection is likely due to an increase in the downstream approximation or estimation error. Table 2 shows that whereas the disparity MAE test loss of $u'$ increases for large depth values, it decreases in the 0-20 meter range. As the pedestrian-detection evaluation focuses on pedestrians within 20 meters, we expect the pedestrian detector $w'$ to improve along with disparity-estimation model $u'$: that is, the upstream error likely decreases.

|  |  | P-RCNN | F-PTNET |
|---|---|---|---|
| | $u$ | 30.53 | 44.10 |
| AP$_{\mathrm{3D}}$ | $u'$ | **34.20** | **46.14** |
| | $y$ | 55.19 | 62.16 |
| | $u$ | 34.09 | 51.84 |
| AP$_{\mathrm{BEV}}$ | $u'$ | **38.83** | **53.99** |
| | $y$ | 64.25 | 65.23 |

Table 3: Average precision at IoU 0.5 of pedestrian detection measured for 3D box view (AP$_{\mathrm{3D}}$) and 2D birds-eye view (AP$_{\mathrm{BEV}}$) for two different detection models (P-RCNN and F-PTNET). Pedestrian detector is trained by removing pedestrians farther than 20 meters. Results are shown for the baseline disparity estimation model, $u$, the improved model, $u'$, and an oracle model, $y$. Higher is better.

However, the point representations produced by model $u'$ are likely noisier for far-away pedestrians than those produced by $u$ (see Table 2). Because the downstream model is trained to (also) detect these far-away pedestrians, the noise in the pedestrian point clouds makes it harder for that model to learn a good model of the shape of pedestrians. This negatively affects the ability of $w'$ to detect nearby pedestrians. Indeed, if the pedestrian detector is only trained on nearby pedestrians, the self-defeating improvement disappears; see Table 3. Hence, the observed self-defeating improvement was due to an increase in a downstream error term: the upstream model did improve for the downstream task, but the downstream model was unable to capitalize on that improvement.

## 5  Related Work

This study is part of a larger body of work around *machine-learning engineering* [9, 21, 23]. Sculley et al. [28] provides a good overview of the problems that machine-learning engineering studies. The self-defeating improvements studied in this paper are due to the entanglement problem [1, 22], where "changing anything changes everything" [28]. Entanglement has also been studied in work on "backward-compatible" learners [29, 32, 35]. Prior work has also studied how the performance of humans operating a machine-learning system may deteriorate when the system is improved [3].

Some aspects of self-defeating improvements have been studied in other domains. In particular, the effect of an upstream model change on a downstream model can be viewed as *domain shift* [4, 25]. However, approaches to domain shift such as importance weighting of training examples [31] are not directly applicable because they require the input space of both models to be identical. The study of self-defeating improvements is also related to the study of *transfer learning*, *e.g.*, [2, 19, 24]. Prior work on transfer learning has reported examples of self-defeating improvements: *e.g.*, [19] reports that some regularization techniques improve the accuracy of convolutional networks on ImageNet, but negatively affect the transfer of those networks to other recognition tasks.

Finally, work on *differentiable programming* [18, 36] is relevant to the study of self-defeating improvements. In some situations, a self-defeating improvement may be resolved by backpropagating learning signals from a model further upstream. However, such an approach is not a panacea as downstream tasks may have conflicting objectives. Moreover, technical or practical obstacles may prevent the use of differentiable programming in practice (see Section 2).

## 6  Conclusion and Future Work

This study explored self-defeating improvements in modular machine-learning systems. We presented a new error decomposition that sheds light on the error sources that can give rise to self-defeating improvements. The findings presented in this study suggest a plethora of directions for future work:

- It may be possible to derive bounds on the error terms in Equation 2 that go beyond the current art. For example, it is well-known that the trade-off between approximation and estimation error leads to an excess risk that scales between the inverse and the inverse square root of the number of training examples [27, 39]. New theory may provide more insight into the effect of the upstream hypothesis set and training set size on those errors, and may bound the upstream error term in 2.
- Our study focused on *first-order* entanglement between two models (either an upstream and a downstream model or two upstream models). Some settings may give rise to higher-order entanglements, for example, between three upstream models whose errors cancel each other out.
- Our definition of different types of entanglement may pave the way for the development of *diagnostic tools* that identify the root cause of a self-defeating improvement.
- We need to develop *best practices* that can help prevent self-defeating improvements from occurring. One way to do so may be by tightening the "API" of the machine-learning models, *e.g.*, by calibrating classifier outputs [17] or whitening feature representations [20]. Such transforms constrain a model's output distribution, which may reduce the downstream negative effects of model changes. Alternatively, we may train models that are inherently more robust to changes in their input distribution, *e.g.*, using positive-congruent training [38] or meta-learning [15].
- We also need to study self-defeating improvements of other aspects. An important open question is [13]: *Can making an upstream model fairer make its downstream dependencies less fair?*

We hope this paper will encourage the study of such questions, and will inspire the machine-learning community to rigorously study machine-learning systems in addition to individual models.

## Acknowledgements

The authors thank Yan Wang, Yurong You, and Ari Morcos for helpful discussions and advice.

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
