# OpenReview forum: "Fixes That Fail: Self-Defeating Improvements in Machine-Learning Systems"
_NeurIPS.cc/2021/Conference — NeurIPS 2021 Poster_

### Official Review · Reviewer_yjBZ · 2021-07-06

**Rating:** 6
**Confidence:** 4

**Summary:**

This paper studies the problem of "self-defeating improvement" (improving an upstream ML model deteriorates a downstream ML model in the same system). Leveraging Bayes error decomposition, it decomposes the error of a system with multiple ML models into: (excess) upstream error, upstream compatibility error (when more than one upstream model is used), downstream approximation error, and downstream estimation error. The paper conducts a case study in Pseudo-LiDAR for vehicle/pedestrian detection to demonstrate the existence of the studied "self-defeating improvement" in real world.

**Limitations And Societal Impact:**

Yes

**Main Review:**

Pros:
- well motivated and important topic
- well-written; intuitive and clean formulation of the problem
- good visualization of the formulation on toy examples
- case study demonstrates the existence of the studied problem in practice

Cons:
- only one case study is conducted
- no systematic method is presented to determine which component of the error is the cause of a certain "self-defeating improvement"

This paper studies an important topic of "self-defeating improvement", which seems to be relatively less formally studied in the field. Overall, I find this paper is a pleasure to read and believe it will encourage more exploration on this important topic.  One minor weakness is that the paper only conducts one case study so I am not sure how common the studied "self-defeating improvement" is in real-world ML systems.

In the case study, the paper analyzes the potential causes for the deterioration of detection for vehicle and pedestrian. Regarding the detection for pedestrian, I wonder if it is possible to distinguish between downstream approximation error and downstream estimation error? Is there a possible systematic way to decompose the error based on the formulation in real-world application like the one in the case study?



Other comments:
- Regarding fairness in the final bullet point in Conclusion and Future Work, it might worth looking into "Fairness Under Composition" by Cynthia Dwork and Christina Ilvento,


Questions:
- Is it possible to also make a visualization for the upstream error similar to Figure 2 and Figure 3?
- For Figure4(a), I am a bit confused here. Are the ground-truth for u, v, and w different? Are you using the same color code here as in Figure2 and Figure3?
----------------------------------------------------------------------------------------------------------------------------------------
The author's response addresses my concerns and I will keep my initial score.


**Time Spent Reviewing:**

3

---

> ### Author Response · Authors · 2021-08-10
>
> Thank you for your insightful comments and positive feedback on our work!
>
> Realistic case study: We have observed self-defeating improvements in multiple machine-learning systems, including the pseudo-LiDAR system. We are unable to publish some of our observations because they were made in proprietary systems, which is why we focus on pseudo-LiDAR in this paper. The pseudo-LiDAR system we use is standard practice in the literature (see [35] and dozens of follow-up papers). Through personal communication, we also know that Tesla is using a similar system to implement their self-driving features. Hence, we believe our case study is a realistic demonstration of self-defeating improvements.
>
> Measuring error terms in case study: Unfortunately, it is impossible to measure exact error terms except in very simple synthetic examples because several variables in the Bayes error decomposition are unknown (for example, we do not know the Bayes classifier for any real system). The same is true for traditional Bayes error decompositions. We emphasize that those error decompositions have, nonetheless, been very useful in understanding and improving machine-learning models. By analogy, we believe our decompositions may help the community understand and improve machine-learning systems, too.
>
> Fairness: Thanks for pointing out this reference! It is, indeed, very relevant to this remark.
>
> Upstream error figure: We had such a figure in early drafts of the paper, but ended up removing it because of space constraints. We will try to include it in the camera-ready version of the paper.
>
> Figure 4a: Apologies for the confusion! Yes the ground-truths for u, v, w are different. In Figure 4, each point in the rectangle is an input x. The colors (red and green) in the three subfigures (from left to right) represent the class labels for the two upstream tasks and the downstream task, respectively. We will clarify this in the camera-ready version of the paper.

---

### Official Review · Reviewer_ARBJ · 2021-07-13

**Rating:** 3
**Confidence:** 3

**Summary:**

The paper asks a question about performance of a system that uses multiple independently trained models, and studies cases where improvements to a model makes overall performance worse (due to errors in other components)

**Limitations And Societal Impact:**

maybe could be more explicit about how the problems mentioned could cause real world harm

**Main Review:**

Overall the paper explores an interesting point, but I'm not sold on the formalism or its connection to the experiments.  The exposition was also not great, IMO.

Section 3: The discussion feels misleading to me.  The claim that improvements to a module are guaranteed to improve the entire system for traditional software systems seems generally wrong.  Even efficiency improvements to software aren't guaranteed to improve the whole system - maybe Cloudflare crashes when another component becomes so fast it dos-es it.  I don't see any clear key distinction between software and ML - the examples in section 4 seem pretty modular.

3.1
Equation 2:  A number of things that would've made this easier to understand:
- It was confusing that w* was described as a downstream model when it doesn't seem to take outputs from an upstream model.
- I would've appreciated a more straightforward pointer or an in-text explanation of the "standard decomposition[s]".
- The notation for optimal model condition on an upstream model feels like it should have a v in it - maybe w*(v) or something.  This seems especially important in the discussion of upstream error.  I can't immediately tell exactly what the equation there is saying (though I have a good guess)
In this section, I also expected a discussion of (v, w) vs. (v', w) and which terms would be expected to increase.  Overall I don't have a clear takeaway of how this decomposition helps clarify much about self-defeating improvements.

3.2

I'm not convinced you need two upstream models for things to be interesting. Couldn't a similar anti-correlation of errors issue occur between an upstream and downstream models?

4

Being unfamiliar with the self-driving field, it's not clear to me whether minimizing depth/disparity MAE and maximizing average precision are aligned objectives.  Is it the case that having optimal models separately for minimizing depth MAE and maximizing average precision would yield optimal performance on the KITTI dataset?  If not, then it seems important to also disentangle the error from poor choice of loss function.  I can't tell that depth MAE isn't just better than disparity MAE and that you didn't simply optimize the wrong (or a wronger) thing.

Overall, I'm not sure I had a clear takeaway from this section, as I think it's fairly obvious one could construct scenarios where this sort of thing happens.  I think the most interesting fact is that it occurred in a realistic setting, which is definitely a point practitioners should definitely be aware of.  I would've liked to see some early steps towards predicting how these errors occur and how to mitigate them.

5

Is backpropagating really fundamentally different at all than just alternating optimization steps between the upstream/downstream models?  It seems like either way you run into local optima issues potentially.

**Time Spent Reviewing:**

1

---

> ### Author Response · Authors · 2021-08-10
>
> Thank you for your insightful comments and constructive feedback on our work!
>
> Comparison with traditional software engineering: The point we are trying to make here is that the goal of traditional software engineering is to compartmentalize a system into modules that can be modified independently: in a well-designed system, changes in one module will not affect the output of another module. Of course, we fully agree that this goal is often not achieved in general, for example, due to leaky abstractions in the system’s design or due to interactions due to modules dispatching computations to the same underlying hardware. We will clarify this in the camera-ready version of the paper.
>
> Equation 2: Thank you for your suggestions on how to clarify the presentation! We will make sure to incorporate them in the camera-ready version of the paper.
>
> Value of error decomposition: We believe that the error decomposition presented in this paper is an important first step towards the goal of understanding and addressing self-defeating improvements, in the same way that traditional Bayes error decompositions have helped the community develop intuitions and best practices for training machine-learning models. For example, traditional error reductions have helped us develop and understand key tools such as regularization and stochastic gradient descent. By analogy, we hope that our error decompositions will help the community understand and improve machine-learning systems.
>
> Design choices in case study: The choice to optimize depth/disparity MAE for stereo and AP for detection is standard practice in the literature on pseudo-LiDAR (see [35] and dozens of follow-up papers). Through personal communication, we also know that Tesla is using a similar system to implement their self-driving features. Hence, we believe it is of interest to show that different types of self-defeating improvements can arise in such systems.
>
> Anti-correlation with single upstream model: It is indeed possible for an upstream and downstream to be anti-correlated, e.g., if the upstream task involves distinguishing cats from dogs and the downstream task is classifying different breeds of dogs. This situation has been studied in prior work (e.g., [18]) and is covered in the upstream error paragraph in section 3.1. When multiple upstream models are involved, there can be interactions between different upstream models in addition to this upstream-downstream anti-correlation. Section 3.2 aims to cover these more complex interactions.
>
> Backpropagation: Backpropagation is somewhat similar to alternating optimization between upstream and downstream models, but target propagation or ADMM-based approaches to training neural networks (for example, Taylor et al. (2016)) would be a better analogy. However, we do not consider alternating optimization in our setting: there is no explicit feedback from the downstream model for how the upstream should update their model. In many practical settings, alternating optimization is infeasible for reasons similar to those outlined in the last paragraph of Section 2.
>
> Reference: Gavin Taylor, Ryan Burmeister, Zheng Xu, Bharat Singh, Ankit Patel, and Tom Goldstein. Training Neural Networks without Gradients: A Scalable ADMM Approach. 2016.

---

> > ### Comment · Reviewer_ARBJ · 2021-08-10
> > **reply**
> >
> > Thanks for your reply, they were helpful context
> >
> > RE Comparison with traditional software engineering:  I'm still not sure I follow, and sorry if I am misunderstanding something here!  My point is that the modules cannot be modified independently if they are all buggy.  When considering "error" for an ML module, it seems more analogous to considering bugs in traditional software than to considering moving between different correct implementations.  Changes in implementation for one module won't affect the output of another module, given the same functionality, but changes in functionality surely do (and i've seen plenty of situations where two bugs "cancel" each other out).  At the end of the day, I probably wouldn't generally endorse the statement "improvements to a module are guaranteed to improve the entire system."

---

> > > ### Author Response · Authors · 2021-08-10
> > >
> > > Large software systems like Google or Facebook are typically organized as a collection of services. These services expose some API and corresponding SLAs (on reliability, efficiency, etc.). If you develop a new service, you only need to know about the API and SLAs of the other services that you are using. In principle, the underlying implementation of one of the services that you use may change and you would not even notice.
> > >
> > > The point we are trying to get at in in the introduction of Section 3 is that, ideally, we would like to get to a similar situation for machine-learning models (services) that are used to compose a larger a system. This is difficult, for example, because there is no SLA on the output distribution of the model or anything like that. The “functionality” of a machine-learning model seems difficult to capture in current APIs and SLAs. We believe that this can contribute to the self-defeating improvements that we analyze in our paper (in addition to other problems [27]).
> > >
> > > But to your point, our current write-up suggests that the current situation in traditional software engineering is one of perfect modularization, which is not quite true in practice. We agree that even in service-based systems, complex interactions between the services can and do arise due to a variety of reasons. The current write-up describes an ideal software-engineering world that probably is not commonly attained in practice, and we should make that much clearer.
> > >
> > > Thank you for pointing out that our write-up is confusing in this regard! We will rewrite this part of Section 3 per this discussion.

---

> > > > ### Comment · Reviewer_ARBJ · 2021-08-10
> > > > **reply**
> > > >
> > > > That makes sense!  Do you think the case study in section 4 is a case of poor modularization though?  Or is it more that there are many errors because the system itself is poor at the task specified?  If it's the latter, I might consider reframing from "ML models don't behave like software modules because they lack an explicit specification for their functionality" to "ML models don't behave like software modules because they are currently very brittle, and fail to satisfy the specifications intended by their creators".

---

> > > > > ### Author Response · Authors · 2021-08-11
> > > > >
> > > > > I suppose it is a little bit of both?
> > > > >
> > > > > If all models in the system had perfect performance, then we wouldn’t have self-defeating improvements: it wouldn’t be possible to improve any of the models in the system, so you naturally wouldn’t have self-defeating improvements either.
> > > > >
> > > > > I think the modularization in the case study in Section 4 is reasonable in the sense that the individual modules solve tasks that are clearly separate and make sense to solve independently. But the modularization is weak in the sense that the modules lack clear specifications. For example, the depth-estimation module could provide additional guarantees on the behavior of the module as part of its SLA: “at depth d, we guarantee that the depth estimate is accurate op to some tolerance 90% of the time” or “the relative error in our depth estimate is constant up to some tolerance across the entire supported depth range”. Providing this guarantee would have forced the developers of the depth-estimation module to be more careful when improving their module, which may have prevented the self-defeating improvement in the detection module.
> > > > >
> > > > > So perhaps the right way to phrase it is: “ML models don't behave like software modules because they are very brittle, and because their creators usually do not provide strict specifications of their behavior”. Would you agree with that phrasing?
> > > > >
> > > > > (Admittedly, a lot more work is needed to figure out what are reasonable SLAs for machine-learning models and how they can be implemented in practice. A simple example could be classifier calibration, which provides a guarantee on the error distribution, but the depth-estimation example may require more complex solutions. We hope that our paper can help inspire work in that direction.)

---

> > > > > > ### Comment · Reviewer_JHen · 2021-08-13
> > > > > > **SW vs ML**
> > > > > >
> > > > > > Although this discussion is interesting, I am not sure there is ultimately a single short way to express the differences.
> > > > > >
> > > > > > A lot of software packages and APIs don't have (formal) specifications and SLAs and break downstream clients/users when changed even when their implementers claim backwards compatibility and/or improvements. A lot of software packages are brittle. So I'd say the “ML models don't behave like software modules because they are very brittle, and because their creators usually do not provide strict specifications of their behavior” does not really capture the situation.
> > > > > >
> > > > > > IMO ML models somewhat differ from SW packages because their spec would likely be a (possibly probabilistic) function in multidimensional space of inputs/outputs that is way more complex than average SW package spec. Although even that is possibly not a real difference if you compare ML model to a software system that controls a power plant...
> > > > > >
> > > > > > So intuitively ML models are different from other SW packages, but that difference might be just because we compare them with some specific subset of SW packages.
> > > > > >
> > > > > > Ultimately, perhaps the right thing to do is not to make very strong claims about difference of ML models and SW packages. Unless you also want to spend time and space arguing that these claims are right (and why).
> > > > > >
> > > > > > Good luck

---

> > > > > > > ### Comment · Reviewer_ARBJ · 2021-08-13
> > > > > > > **agree with JHen**
> > > > > > >
> > > > > > > I agree with JHen, the situation seems a lot more subtle
> > > > > > >
> > > > > > > Arguably the ideal specification for a cat vs. dog classifier is "when humans would agree the image is obviously a cat, it says cat.  when humans would agree the image is obviously a dog, it says dog".  But
> > > > > > > -  this is pushing all the difficulty/fuzziness into the "when humans would agree" condition; traditional software would never have a spec like that.  it's quite plausible to me the very kinds of problems ML tackles that software cannot are the ones which aren't as decomposable
> > > > > > > -  a model is unlikely to meet that spec, and so may have a super mediocre (and also fuzzy) SLA of the form "when the image is not too out of distribution, it has a 90% chance of meeting the spec"

---

> > > > > > > > ### Author Response · Authors · 2021-08-15
> > > > > > > >
> > > > > > > > Thanks for a very interesting and valuable discussion!
> > > > > > > >
> > > > > > > > I agree with the conclusion here: we should not explicitly contrast machine-learning systems with other software systems because there is large variation in the degree of modularization that those software systems exhibit, which makes the comparison confusing. Moreover, the comparison with traditional software systems appears unnecessary because it seems orthogonal to our analyses and observations on self-defeating improvements in machine-learning systems that our paper makes.
> > > > > > > >
> > > > > > > > We will update the relevant paragraph accordingly. Thanks a ton for helping us to improve our manuscript!

---

> > ### Author Response · Authors · 2021-08-19
> > **Has our response addressed your concerns?**
> >
> > Hello reviewer ARBJ: We would be grateful if you can confirm whether our response and subsequent discussion has addressed some of the concerns you raised in your review? Please let us know if any issues remain and/or if there are any additional clarifications we can provide.

---

### Official Review · Reviewer_JHen · 2021-07-16

**Rating:** 5
**Confidence:** 4

**Summary:**

Authors present evidence that improvements in models that are components of larger system may lead to worse results for the whole system or downstream models.

**Limitations And Societal Impact:**

Authors present the limitations of their work. The major limitation is that they do not propose solutions, just identify and present the problem. Although presenting the problem is useful, the presentation is not particularly novel.

Authors mention some societal impacts related to their work, i.e. possibility of self-defeating improvements in bias (i.e. improving bias of subalgorithm possibly making system more biased). This is worthwhile societal observation.

**Main Review:**

The paper shows that in a system with multiple machine learning models improving some models may lead to deterioration of results of the whole system or models dependent on improved models.

Authors present problem and some formalization of it. They do not suggest solutions for the problem. Authors also show , illustrative examples and examples from real world.

I believe that the issue of "self-defeating improvements" is known. Authors do an adequate job of presenting it in general and via examples.

The formalization of problem seems to be overly complicated perhaps by the choice of math notation. It is not clear that the formalization as presented provides much benefit. This is possibly made worse by authors' choices in the notation.

Please do not use the same letters (u, v, w) to denote concrete nodes in the graph (see Figure 1) and as abstract variables in formulas. This "reuse" of the letters makes the text unnecessarily hard to read, because reader has to continuously check if you are referring to an abstract variable or to the concrete node in the graph.

It seems that the notation in (2) is not the same as notation in (3). Overall notation seems to be non-intuitive. For example, why name "optimal model conditioned on an upstream model" w-dagger instead of showing the conditioning via subscript or superscript? Especially since in (3) you use w-dagger-with-u,v-subscript? The notation can be better thought out so it would be more readable and understandable.

What does "anti-correlated" mean? Is this defined mathematically or just an informal expression?

Line 84: what do you mean that test set of downstream model is fixed even though inputs of it may change. If inputs change, how can you say that the test set is fixed?

Line 212: I don't think you can say that "optimal v depends on u". The right expression might be that "v output that is beneficial to w depends on u". Or you should distinguish v optimality in general vs v optimality for w given u.



**Time Spent Reviewing:**

3.5

---

> ### Author Response · Authors · 2021-08-10
>
> Thank you for your insightful comments and positive feedback on our work!
>
> Solutions: Of course, the final goal of this line of work is to develop solutions and best practices that allow machine-learning practitioners to deploy systems that are less susceptible to self-defeating improvements. We believe that the analysis and error decomposition presented in this paper are an important first step towards that goal, in the same way that traditional Bayes error reductions have helped the community develop intuitions and best practices for machine learning more generally.
>
> Formalization and notation: Thank you for your suggestions on how to clarify the problem formalization and improve the notation! We will make sure to incorporate them in the camera-ready version of the paper.
>
> Use of term “anti-correlated”: We used the term “anti-correlated error” informally to intuitively capture the statistical notion of anti-correlation. We opted to not formalize the notion as it would further complicate the notation without providing much additional insight.
>
> Fixed test set: The examples that are input into the machine-learning system (that is, the input into the graph) to test its performance are fixed and do not change. The motivation for this is that we do not want to move the goalposts in between system evaluations. Of course, the input into individual models can change because that input is determined in part by its ascendants in the graph (which may change in between evaluations). We will clarify this in the camera-ready version of the paper.
>
> Optimal v depends on u: Thanks for pointing this out! We intend to distinguish v-optimality in general versus v-optimality for w, but we agree the current phrasing here is unclear. We will clarify this in the camera-ready version of the paper.

---

> > ### Comment · Reviewer_JHen · 2021-08-24
> > **Thank you for your response**
> >
> > Thank you for your response.

---

### Official Review · Reviewer_QoL7 · 2021-07-16

**Rating:** 7
**Confidence:** 4

**Summary:**

This paper investigates a practical problem referred to as self-defeating improvements when developing machine learning systems for real world applications such as self-driving cars or virtual assistants. That is, given a machine learning system with multiple components depending on each other, developing one component independently could sometimes make the overall system worse. This work first studies the problem in a system with two models (upstream task and downstream task) and decomposes the error into upstream error, downstream approximation error, and downstream estimation error. Building on top of the formulation, this work expands the discussion on a system with two or more upstream models. Finally, it demonstrates the emergence of self-defeating improvements in a realistic pseudo-LiDAR detection system.

**Ethical Concerns:**

N / A

**Limitations And Societal Impact:**

Yes

**Main Review:**

Strengths:
* [S1] This paper studies a very important problem in developing machine learning systems in the real world with several dependent modules. This work has a clear definition and formulation of the problem.
* [S2] This work decomposes the self-defeating error into three components and provides analysis for each of them. The reviewer finds figure 2-4 very illustrative.
* [S3] This work also demonstrates the real case in the Pseudo-LiDAR platform for object detection. This shows the real-world impact of the problem this paper studies.

Weaknesses:
* [W1] The actual impact is still limited as this work does not provide a solution to the problem.
* [W2] While the error decomposition is interesting, the reviewer would expect to see such decomposition in the case study. Is it possible to quantify the error decomposition in the experiments? Please elaborate on this in the rebuttal.

**Time Spent Reviewing:**

8

---

> ### Author Response · Authors · 2021-08-10
>
> Thank you for your insightful comments and positive feedback on our work!
>
> Solutions: Of course, the final goal of this line of work is to develop solutions and best practices that allow machine-learning practitioners to deploy systems that are less susceptible to self-defeating improvements. We believe that the analysis and error decomposition presented in this paper are an important first step towards that goal, in the same way that traditional Bayes error reductions have helped the community develop intuitions and best practices for machine learning more generally.
>
> Error decomposition: The experimental results in the case study suggest different error components increasing or decreasing. For example, the self-defeating improvement in Table 2 is due to an increase in upstream error, whereas the self-defeating improvement in Table 3 is caused by the downstream error term increasing. Having said that, it is impossible to measure the exact error terms except in very simple synthetic examples because several variables in the error decomposition are unknown (for example, we do not know the Bayes classifier in the pseudo-LiDAR case study). The same is true for traditional Bayes error decompositions. We emphasize that those error decompositions have, nonetheless, been very useful in understanding and improving machine-learning models. By analogy, we believe our decompositions may help the community understand and improve machine-learning systems, too.

---

### Official Review · Reviewer_RhPc · 2021-07-16

**Rating:** 7
**Confidence:** 3

**Summary:**

This paper explores, "self-defeating improvements, potential negative
effects of improving a part of a module machine learning system. The
causes of such self-defeating improvements are analyzed through Bayes
error decomposition. The paper presents a case study with pedestrian
and car detection where both tasks rely on an upstream disparity
estimation model. The results show that despite improvements in
disparity estimation (when tested alone), the overall system
performance degrades.


**Limitations And Societal Impact:**

I think the limitations of the work are adequately stated in the paper.

**Main Review:**

The paper explores a relatively less studied problem which will be
more relevant as the number of more complex (and modular) machine
learning systems increase. The main contribution  of the current paper
is demonstrating the problem, and providing an analysis of sources of
the self-defeating improvements.

The following of some points (roughly ordered by importance) that I
miss or think would be improve the paper.

- I think raising awareness and promoting discussion are reasonable
  contributions. However, paper would be stronger if included some
  concrete discussion towards solutions (e.g., diagnostics for
  detecting the problems, or suggestions to avoid them).
- It is not clear to me whether the case study is a constructed
  demonstration, or an example that the authors happened to observe in
  a real study.
- Although the authors state in the checklist that they did not report
  variation (error bars) because it was low, I think it would still
  help reader decide the extent of the differences if the variation
  was presented in Table 1 and Table 3. Some of the scores in Table 1
  are particularly close.
- The paper is rather dense. In particular, the description of the
  case study is difficult to understand people outside the field
  without consulting quite a few additional sources. I think it is
  important for the current paper to aim for a broader audience.
- Part of the "motivation and problem setting" subsection (line 102)
  is already stated in the introduction, which, I think, is a better
  place for motivation anyway.
- Footnote marks should follow punctuation.


**Time Spent Reviewing:**

6-8

---

> ### Author Response · Authors · 2021-08-10
>
> Thank you for your insightful comments and positive feedback on our work!
>
> Solutions: Of course, the final goal of this line of work is to develop solutions and best practices that allow machine-learning practitioners to deploy systems that are less susceptible to self-defeating improvements. We believe that the analysis and error decomposition presented in this paper are an important first step towards that goal, in the same way that traditional Bayes error reductions have helped the community develop intuitions and best practices for machine learning more generally.
>
> Realistic case study: We have observed self-defeating improvements in multiple machine-learning systems, including the pseudo-LiDAR system. We are unable to publish some of our observations because they were made in proprietary systems, which is why we focus on pseudo-LiDAR in this paper. The pseudo-LiDAR system we use is standard practice in the literature (see [35] and dozens of follow-up papers). Through personal communication, we also know that Tesla is using a similar system to implement their self-driving features. Hence, we believe our case study is a realistic demonstration of self-defeating improvements.
>
> Variance in results: The standard deviation between runs in Table 1 and 3 is approximately 0.9 AP. We will include all standard deviations in the camera-ready version of the paper.
>
> Suggestions on presentation: Thank you for these suggestions! We will make sure to incorporate them in the camera-ready version of the paper.

---

### Author Response · Authors · 2021-08-10

We thank all the reviewers for their insightful comments and their positive evaluation of our work! We are delighted to see that the reviewers recognize the importance of studying self-defeating improvements in machine-learning systems, and that they recognize the value of our case study and novel Bayes error decompositions.

Several reviewers pointed out that, while our paper formalizes the problem of self-defeating improvements and presents new analyses and error decompositions for this problem, it does not present concrete solutions to the problem. Of course, the final goal of this line of work is to develop solutions and best practices that allow machine-learning practitioners to deploy systems that are less susceptible to self-defeating improvements. We believe that the analysis and error decomposition presented in this paper are an important first step towards that goal, in the same way that traditional Bayes error decompositions have helped the community develop intuitions and best practices for training machine-learning models. For example, traditional error reductions have helped us develop and understand key tools such as regularization and stochastic gradient descent. By analogy, we hope that our error decompositions will help the community understand and improve machine-learning systems.

As a concrete example, the case study on pedestrian detection in pseudo-LiDAR illustrates how the error decomposition can be used to diagnose self-defeating improvements. Intuitively, updating the upstream model to optimize the disparity MAE loss should only improve downstream detection performance as most pedestrians are within the 0-20 meter range. This intuition can be confirmed by Table 2. As a result, our error decomposition suggests that upstream error has not increased, hence it is more likely that the downstream model is not optimized for the updated upstream model, increasing the approximation and/or estimation error terms. Such an analysis follows naturally from the error decomposition framework, but is not straightforward to derive otherwise.

We view our paper as the start of a journey rather than as the end of it: for example, we have ideas on potential solutions (see Section 6) that we are currently investigating. More importantly, we hope that our paper will inspire the community to help us study self-defeating improvements!

---

### Decision · Program_Chairs · 2021-09-27

**Decision:**

Accept (Poster)

**Comment:**

Three reviewers gave favorable scores, one borderline, and one negative (ARBJ). The last reviewer engaged in a productive discussion with the authors, so we can expect the final paper to be improved. The paper addresses a important question and has been refined a lot since it was first submitted to ICML 2021, so it deserves to be published now.

The paper addresses a problem that is important and will be of interest to a broad audience. The issue is that in a system that uses multiple machine learning models, improving the accuracy of some component models may lead to worse results for the system as a whole. The paper explains the problem and some formalizations of it. Although solutions are nor provided, illustrative examples and cases from the real world are described.